# The complete chloroplast genome sequences of six *Hylotelephium* species: Comparative genomic analysis and phylogenetic relationships

Sung-Mo An[1]☯, Bo-Yun Kim[2]☯, Halam Kang[1], Ha-Rim Lee[1], Yoo-Bin Lee[1], Yoo-Jung Park[1], Kyeong-Sik Cheon[1]*, Kyung-Ah Kim[3]*

1 Department of Biological Science, Sangji University, Wonju, South Korea, 2 Plant Resources Division, National Institute of Biological Resources, Incheon, South Korea, 3 Environmental Research Institute, Kangwon National University, Chuncheon, South Korea

☯ These authors contributed equally to this work.
* cheonks@sangji.ac.kr (KSC); 1seizetheday@kangwon.ac.kr (KAK)

**Data Availability Statement:** All complete chloroplast genome sequences are available from the NCBI data base (http://www.ncbi.nlm.nih.gov/)

## Abstract

To evaluate the phylogenetic relationships between *Hylotelephium* and *Orostachys*, and to provide important information for further studies, we analyzed the complete chloroplast genomes of six *Hylotelephium* species and compared the sequences to those of published chloroplast genomes of congeneric species and species of the closely related genus, *Orostachys*. The total chloroplast genome length of nineteen species, including the six *Hylotelephium* species analyzed in this study and the thirteen *Hylotelephium* and *Orostachys* species analyzed in previous studies, ranged from 150,369 bp (*O. minuta*) to 151,739 bp (*H. spectabile*). Their overall GC contents were almost identical (37.7–37.8%). The chloroplast genomes of the nineteen species contained 113 unique genes comprising 79 protein-coding genes (PCGs), 30 transfer RNA genes (tRNAs), and four ribosomal RNA genes (rRNAs). Among the annotated genes, fourteen genes contained one intron, and two genes contained two introns. The chloroplast genomes of the nineteen *Hylotelephium* and *Orostachys* species had identical structures. Additionally, the large single copy (LSC), inverted repeat (IR), and small single copy (SSC) junction regions were conserved in the *Hylotelephium* and *Orostachys* species. The nucleotide diversity between the *Hylotelephium* chloroplast genomes was extremely low in all regions, and only one region showed a high Pi value (>0.03). In all nineteen chloroplast genomes, six regions had a high Pi value (>0.03). The phylogenetic analysis showed that the genus delimitation could not be clearly observed even in this study because *Hylotelephium* formed a paraphyly with subsect. *Orostachys* of the genus *Orostachys*. Additionally, the data supported the taxonomic position of *Sedum taqeutii*, which was treated as a synonym for *H. viridescens* in previous studies, as an independent taxon.

under accession numbers MZ519882, OP537241, OP537242, OP537243, OP537245, OK094424 and MK341052.

**Funding:** This paper was supported by the National Institute of Biological Resources (NIBR) [No. NIBR202104101] received by KSC, and the Graduate School of Sangji University'.

**Competing interests:** The authors have declared that no competing interests exist.

## Introduction

The family Crassulaceae DC. consists of approximately 1500 species in 35 genera that are mainly distributed throughout the Northern Hemisphere [1–3]. Among these, the genus *Hylotelephum* H. Ohba includes approximately 28 taxa that are distributed in Asia, Europe, and North America [3].

The taxa belonging to this genus are perennial herbaceous plants that are usually succulent. The morphological characteristics of this genus are as follows: the roots are fibrous or tuberous and often carrot shaped. The rhizomes are short and fleshy or woody. The stems are erect or decumbent and green or red. The young branches are not covered with scales. The stem leaves are alternate, opposite or 3-5-verticillate, and lanceolate, ovate, orbicular, or oblong; the blade is flat and glabrous, and the margin is serrate. The inflorescence is terminal and sometimes also subterminal and paniculate, cyme, often corymbiform, and sometimes umbel-like. The flowers are bisexual (rarely unisexual), subsessile or pedicellate, and pentamerous. The sepals are usually shorter than the petals. The petals are subconnate at the base and are purple, red, pink, white, occasionally yellowish, or greenish. The seeds are elliptical and have narrow wings [4, 5].

This genus was initially described as *Sedum* L. by Linne [6], and then Miller [7] and Hill [8] proposed the classification of *Hylotelephum* within the genera *Anacampseros* L. and genus *Telephium* L, respectively. Gray [9] recognized this genus as a section of *Sedum*, and many taxonomists [2, 10–12] agreed with Gray's opinions, but Clausen [13] disagreed with this and classified it as a subgenus of *Sedum*. In addition, Ohba [14] recognized it as an independent genus because *Hylotelephum* differs from the other species of *Sedum* by having stipitate or attenuated ovaries, flat broad leaves, compound corymbose inflorescences, and nonyellow petals. However, Chung & Kim [15] proposed the classification of this genus as a section within *Sedum*.

Since then, various molecular phylogenetic studies [16–23] have been performed and revealed a large phylogenetic distance between the two genera; *Hylotelephium* formed a clade with *Sinocrassula* A. Berger, *Orostachys*, and *Meterostachys* Nakai (*Telephium* clade), and *Sedum* was in the Acre clade. However, in most studies, including studies using ITS of nuclear DNA [16, 18] and some marker regions of the chloroplast genome [19], as well as whole chloroplast genome sequences [23], the taxonomic position of *Hylotelephium* as an independent genus was not supported because it formed a paraphyletic clade with *Orostachys* species. Therefore, studies are needed to evaluate an accurate taxonomic position by identifying the exact phylogenetic relationship between two genera.

Furthermore, it is known as a very difficult group to classify [15, 24] because members of *Hylotelephium* have very similar external morphological characteristics between species, and each taxon has wide variation in external morphological characteristics. Because of this, the identities of many taxa are unclear, and scientific names have been repeatedly mixed and misused.

With the development of next-generation sequencing (NGS) technology that has reduced the time and cost required for sequencing, many studies have performed whole chloroplast genome sequencing. These studies have provided much information about plant systematics and evolution. The rapidly evolving loci identified by these studies are very important for resolving unclear phylogenetic relationships because they have higher resolution than traditional molecular markers [25, 26]. Therefore, many studies have focused on finding genetic regions among specific families or genera to provide useful information about molecular markers for further studies [25–32].

We obtained the whole chloroplast genome sequences of six *Hylotelephium* species (*H. pallescens* (Freyn) H. Ohba, *H. spectabile* (Boreau) H. Ohba, *H. ussuriense* (Kom.) H. Ohba,

*H. viridenscens* (Nakai) H. Ohba, *H. viviparum* (Maxim.) H. Ohba and *S. taquetii* Praeger, and compared the sequence to those of thirteen published congeneric and closely related genera (*Orostachys*) chloroplast genomes, i.e., those from *H. erythrostictum* (Miq.) H. Ohba, *H. ewersii* (Ledebour) H. Ohba, *H. verticillatum* (L.) H. Ohba, *O. chongsunensis* Y.N. Lee, *O. iwarenge* (Makino) H. Hara, *O. iwarenge* f. *magnus* Y.N. Lee, *O. japonica* (Maxim.) A. Berger, *O. japonica* f. *polycephala* (Makino) H. Ohba, *O. latielliptica* Y.N. Lee, *O. malacophylla* (Pall.) Fisch, *O. margaritifolia* Y.N. Lee, *O. minuta* (Kom.) Berger. and *O. ramosa* Y.N. Lee. Our main goal was (1) to evaluate the phylogenetic relationships between *Hylotelephium* and *Orostachys* at the whole chloroplast genome level and (2) to identify the taxonomic position of *S. taquetii*, which was treated as a synonym of *H. viridescens* by Ohba. Furthermore, we also aimed (3) to provide important information about the most suitable chloroplast molecular markers for further studies to solve unclear phylogenetic relationships of *Hylotelephium*.

## Materials and methods

### Taxon sampling, DNA extraction, and sequencing

Since the six *Hylotelephium* taxa examined in this study were not endangered or protected species, plant materials were collected without permission. The plant materials for this study were collected from the native habitats of each taxon, and the voucher specimens were deposited in the Sangji University Herbarium (SJUH) (S1 Table). Total DNA was extracted from approximately 100 mg of fresh leaves using a DNeasy Plant Mini Kit (Qiagen Inc., Valencia, CA, USA), and LabChip GXII (PerkinElmer, Inc., MA, USA) was used to quantify the DNA concentration and quality. Genomic libraries were prepared using a TruSeq DNA Sample Preparation Kit (Illumina Inc., San Diego, CA, USA) and paired-end sequencing was performed on a MiSeq platform at LabGenomics, (Seongnam, Korea). The DNA of the *Hylotelephium* taxa was sequenced to produce 3,750,814–4,573,271 raw reads with lengths of 301 bp (S1 Table).

### Assembly and annotation

Low-quality sequences (Phred score < 20) were trimmed using CLC Genomics Workbench (version 6.04; CLC Inc., Arhus, Denmark). Then, *de novo* assembly was implemented using the Geneious assembler with a medium sensitivity option via Geneious Prime v.2022.1.1 (Biomatters Ltd., Auckland, New Zealand). A total of 124,975–421,454 reads were aligned (S1 Table) and selected from chloroplast contigs using Geneious Prime v.2022.1.1. The draft genome contigs were merged into a single contig by joining the overlapping terminal sequences of each contig. The protein-coding genes (PCGs), transfer RNAs (tRNAs), and ribosomal RNAs (rRNAs) in the chloroplast genome were predicted and annotated using Geneious Prime v.2022.1.1 and manually edited by comparison with the published chloroplast genome sequences of *Hylotelephium*. The tRNAs were confirmed using tRNAscan-SE [33]. A circular chloroplast genome map was drawn using the OGDRAW program [34].

### Comparative analyses in *Hylotelephium* and allied genera

The newly complete chloroplast genome sequences of six *Hylotelephium* taxa were used along with the following chloroplast genome sequences from GenBank of NCBI for comparative analysis: three published *Hylotelephium*, *H. ewersii* (MN794014), *H. erythrostictum* (MZ519882), and *H. verticillatum* (MT558730); and ten *Orostachys*, *O. choungsunensis* (ON979333), *O. iwarenge* (ON979332), *O. iwarenge* f. *magnus* (MW851201), *O. japonica* (MW579549), *O. japonica* f. *polycephala* (ON979327), *O. latielliptica* (ON979328),

*O. malacophylla* (ON979331), *O. margaritifolia* (ON979329), *O. minuta* (OK094425) and *O. ramose* (ON979330).

Relative synonymous codon usage (RSCU) and amino acid frequency in the protein coding gene region were determined by DnaSP 6 [35]. The genome structures of the nine species were compared using the MAUVE program [36]. Additionally, the program mVISTA was used to compare similarities between the nine species using the shuffle-LAGAN mode [37]. The annotated *H. verticillatum* chloroplast genome was used as a reference. The large single copy/ inverted repeat (LSC/IR) and inverted repeat/small single copy (IR/SSC) boundaries of these species were also compared and analyzed.

### Nucleotide diversity and repeat analysis

To assess the nucleotide diversity (Pi) between the nineteen chloroplast genomes, including nine *Hylotelephium* and ten *Orostachys* genomes, the complete chloroplast genome sequences were aligned using the MAFFT [38] aligner tool and manually adjusted with BioEdit [39]. We then performed sliding window analysis to calculate the nucleotide variability (Pi) values using DnaSP 6 [35] with a window length of 600 bp and a step size of 200 bp [40].

The REPuter program [41] was used to identify repeats: forward, reverse, palindrome, and complement sequences. The following settings for repeat identification were used: (1) Hamming distance equal to 3; (2) minimal repeat size set to 30 bp; and (3) maximum computed repeats set to 90 bp. The simple sequence repeats (SSRs) were identified by MISA software (http://pgrc.ipk-gatersleben.de/misa/) with the parameters set as follows: 10 for mono-, 5 for di-, 4 for tri-, and 3 for tetra-, penta-, and hexanucleotides [42].

### Phylogenetic analysis

The whole chloroplast genome sequences from 41 Crassulaceae species were compiled into a single file of size 165,758 bp and aligned using MAFFT [37]. Thirty-nine Telephium clade [19] species were selected as the ingroups, and two species from subfam. Kalachoideae (*Cotyledon tomentosa* Harv. and *Kalanchoe delagoensis* Eckl. & Zeyh.) were chosen as the outgroups (S2 Table). Maximum likelihood (ML) analyses were performed using raxmlGUI v.2.0.6 with 1000 bootstrap replicates and the GTR+I+Γ model [43]. Bayesian inference (ngen = 1,000,000, samplefreq = 200, burninfrac = 0.25) was carried out using MrBayes v3.0b3 [44], and the best substitution model (GTR+I+Γ) was determined by the Akaike information criterion (AIC) in jModeltest version 2.1.10 [45].

## Results

### Chloroplast genome features

The chloroplast genomes of six *Hylotelephium* species have been submitted to GenBank of the National Center for Biotechnology Information (NCBI) (Table 1 and S1 Table). The total length of the chloroplast genomes of the nineteen species, i.e., the six *Hylotelephium* species analyzed in this study and the thirteen species analyzed in previous studies, ranged from 150,369 bp (*O. minuta*) to 151,739 bp (*H. spectabile*). Among the *Hylotelephium* species, *H. viviparum* was the smallest (150,430 bp) (Table 1 and Fig 1). All nineteen chloroplast genomes exhibited a typical quadripartite structure, including a large single copy (LSC) region, a small single copy (SSC) region and a pair of inverted repeat (IR) regions. The length of the LSC region ranged between 81,991–83,252 bp, and the GC content of the LSC regions was similar in nineteen species, ranging from 35.7–35.9%. The length of the SSC region was distributed between 16,702 and 17,808 bp, with a GC content of 31.5–31.8%. The length range of the IR

**Table 1. Comparison of chloroplast genome features of *Hylotelephium* and *Orostachys*.**

| Taxa | Length (bp) | | | | %GC | No. of genes | | | | Accession No. |
|---|---|---|---|---|---|---|---|---|---|---|
| | Total | LSC | SSC | IR | | Total | PCG | tRNA | rRNA | |
| *H. erythrostictum* | 151,707 | 83,037 | 16,702 | 25,984 | 37.8 | 113 | 79 | 30 | 4 | MZ519882 |
| *H. ewersii* | 151,699 | 83,253 | 16,838 | 25,804 | 37.7 | 113 | 79 | 30 | 4 | MN794014 |
| *H. pallescens* | 151,717 | 83,236 | 16,879 | 25,801 | 37.8 | 113 | 79 | 30 | 4 | OP537241 |
| *H. spectabile* | 151,793 | 83,105 | 17,080 | 25,804 | 37.8 | 113 | 79 | 30 | 4 | OP537242 |
| *H. ussuriense* | 151,329 | 82,929 | 16,808 | 25,796 | 37.8 | 113 | 79 | 30 | 4 | OP537243 |
| *H. verticillatum* | 151,398 | 82,951 | 16,839 | 25,804 | 37.8 | 113 | 79 | 30 | 4 | MT558730 |
| *H. viridescens* | 151,650 | 83,175 | 16,873 | 25,801 | 37.8 | 113 | 79 | 30 | 4 | OK094424 |
| *H. viviparum* | 150,430 | 81,991 | 16,833 | 25,803 | 37.8 | 113 | 79 | 30 | 4 | OK094424 |
| *S. taquetii* | 151,650 | 83,175 | 16,873 | 25,801 | 37.8 | 113 | 79 | 30 | 4 | OP537245 |
| *O. chongsunensis* | 151,399 | 82,898 | 16,875 | 25,813 | 37.8 | 113 | 79 | 30 | 4 | ON979333 |
| *O. iwarenge* | 151,431 | 82,924 | 16,881 | 25,813 | 37.8 | 113 | 79 | 30 | 4 | ON979332 |
| *O. iwarenge f. magnus* | 151,276 | 82,784 | 16,868 | 25,812 | 37.8 | 113 | 79 | 30 | 4 | MW851201 |
| *O. japonica* | 150,464 | 83,035 | 16,859 | 25,285 | 37.7 | 113 | 79 | 30 | 4 | MW579549 |
| *O. japonica f. polycephala* | 150,464 | 83,035 | 16,859 | 25,285 | 37.7 | 113 | 79 | 30 | 4 | ON979327 |
| *O. latielliptica* | 151,462 | 83,004 | 16,866 | 25,796 | 37.7 | 113 | 79 | 30 | 4 | ON979328 |
| *O. malacophylla* | 151,374 | 82,872 | 16,876 | 25,813 | 37.8 | 113 | 79 | 30 | 4 | ON979331 |
| *O. margaritifolia* | 151,112 | 82,562 | 16,842 | 25,854 | 37.8 | 113 | 79 | 30 | 4 | ON979329 |
| *O. minuta* | 150,369 | 82,795 | 16,854 | 25,360 | 37.7 | 113 | 79 | 30 | 4 | OK094425 |
| *O. ramosa* | 151,424 | 82,924 | 16,874 | 25,813 | 37.8 | 113 | 79 | 30 | 4 | ON979330 |

region of nineteen species was 25,285–25984 bp, which contained 42.8–43.0% GC content. The chloroplast genomes of the nineteen species contained 113 unique genes comprising 79 PCGs, 30 tRNAs, and four rRNAs (Table 1 and S3 Table). In addition, nineteen genes, including eight protein-coding genes (*rpl23*, *ycf2*, *ndhB*, *rps7*, 3'-end *rps12*, and part of *rps19* and *ycf1*), seven tRNAs (*trnI-CAU*, *trnL-CAA*, *trnV-GAC*, *trnI-GAU*, *trnA-UGC*, *trnR-ACG* and *trnN-GUU*) and four rRNAs (*16S rRNA*, *23S rRNA*, *4.5S rRNA* and *5S rRNA*), were duplicated in IR regions. The *rps12* gene had trans-splicing, and its 3'-end was duplicated in the IR region, while its 5'-end was present in the LSC region (Fig 1). Among the annotated genes, fourteen genes (*atpF*, *ndhA*, *ndhB*, *petB*, *petD*, *rpl2*, *rps12*, *rpl16*, *rpoC1*, *trnA-UGC*, *trnI-GAU*, *trnK-UUU*, *trnL-UAA*, and *trnV-UAC*) contained one intron, and two genes (*clpP* and *ycf3*) contained two introns (S3 Table).

A total of 23,968–26,063 codons were identified in nine *Hylotelephium* species (S4 Table). Among them, AUU (4.1–4.2%, isoleucine), AAA (4.1%, lysine), and GAA (3.9%, glutamic acid) were the most frequently used codons, while CGC (0.4–0.5, arginine) and UGC (0.3%, cysteine) had the lowest usage rates. There were 29 codons with relative synonymous codon usage (RSCU) values greater than 1, 2 of which were equal to 1, and 30 were less than 1. In addition, the codons containing A or T at the 3'-end mostly had RSCU > 1 (26 out of 29 codons), and most of the codons containing G or C at the 3'-end were less than 1 or equal to 1 (28 out of 30 codons).

The results of the chloroplast genome structure comparison between the nineteen *Hylotelephium* and *Orostachys* species using MAUVE [35] showed that all chloroplast genomes were the same (S1 Fig). The pairwise cp genomic alignments between all 19 *Hylotelephium* and *Orostachys* species showed very high similarity in all sequences (Fig 2). The LSC and SSC regions were more variable than the IR regions, and noncoding regions were more susceptible to mutations than coding regions. In addition, *clpP*, *ndhA* and *ycf1* were the most different

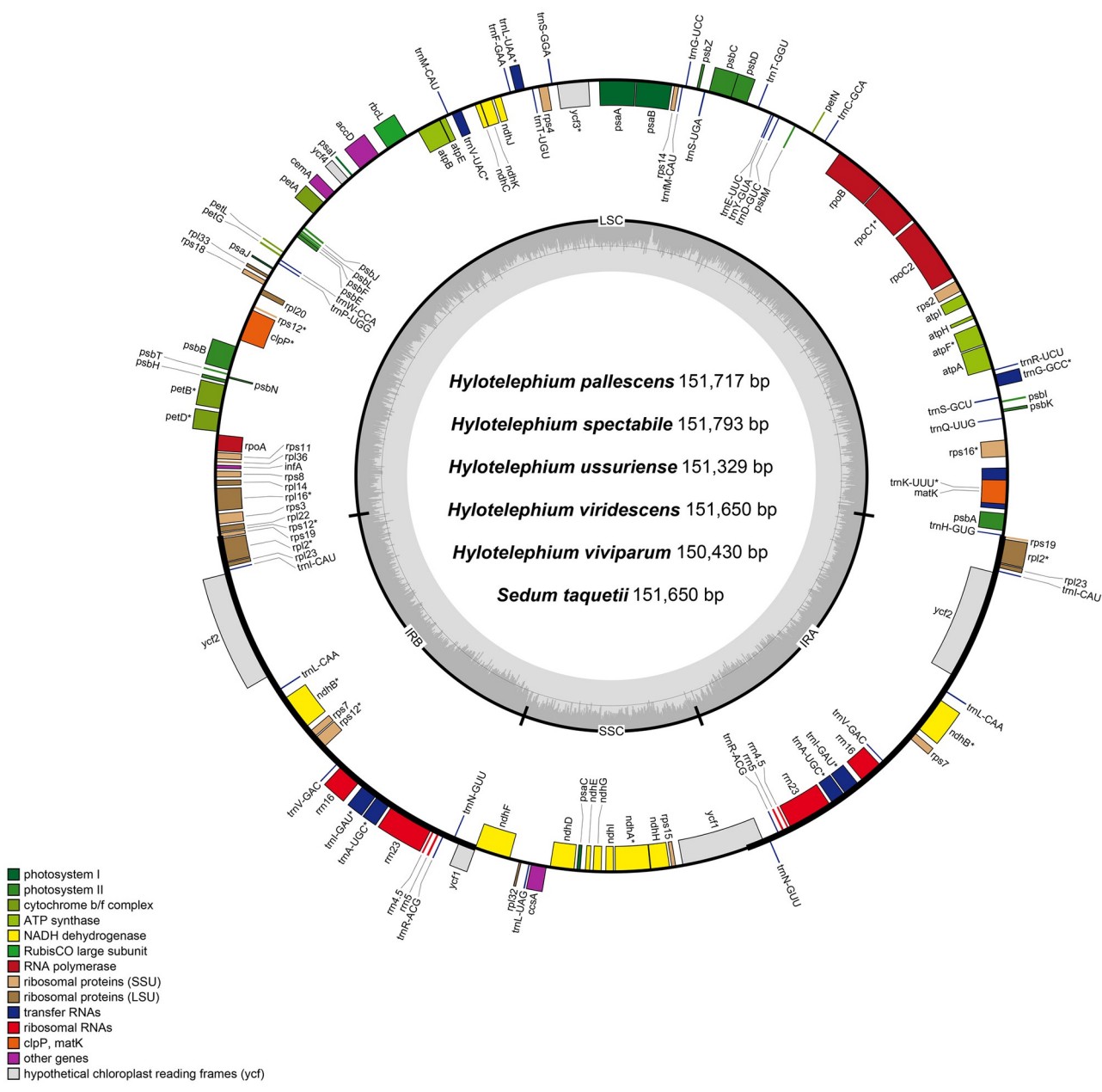

**Fig 1. Map of the newly analyzed chloroplast genome of six *Hylotelephium* species.** Genes inside the circle are transcribed clockwise, and genes outside are transcribed counterclockwise. The dark gray inner circle corresponds to the GC content, and the light gray circle corresponds to the AT content.

from each other among the coding regions, and in the noncoding regions, *rps16-trnQ(UUG)*, *atpH-atpI*, *trnE(UUC)-trnT(GGU)*, *psbZ-trnG(UCC)*, *ycf4-cemA* and *ycf2-trnL(CAA)* of were quite different from each other.

The border regions and adjacent genes of chloroplast genomes were compared to analyze the expansion and contraction variation in junction regions, which are common phenomena in the evolutionary history of terrestrial plants [46]. A comparison of the LSC/IR and IR/SSC boundaries in the nineteen species is shown in Fig 3. The *rps19, ndhF* and *ycf1* genes of all

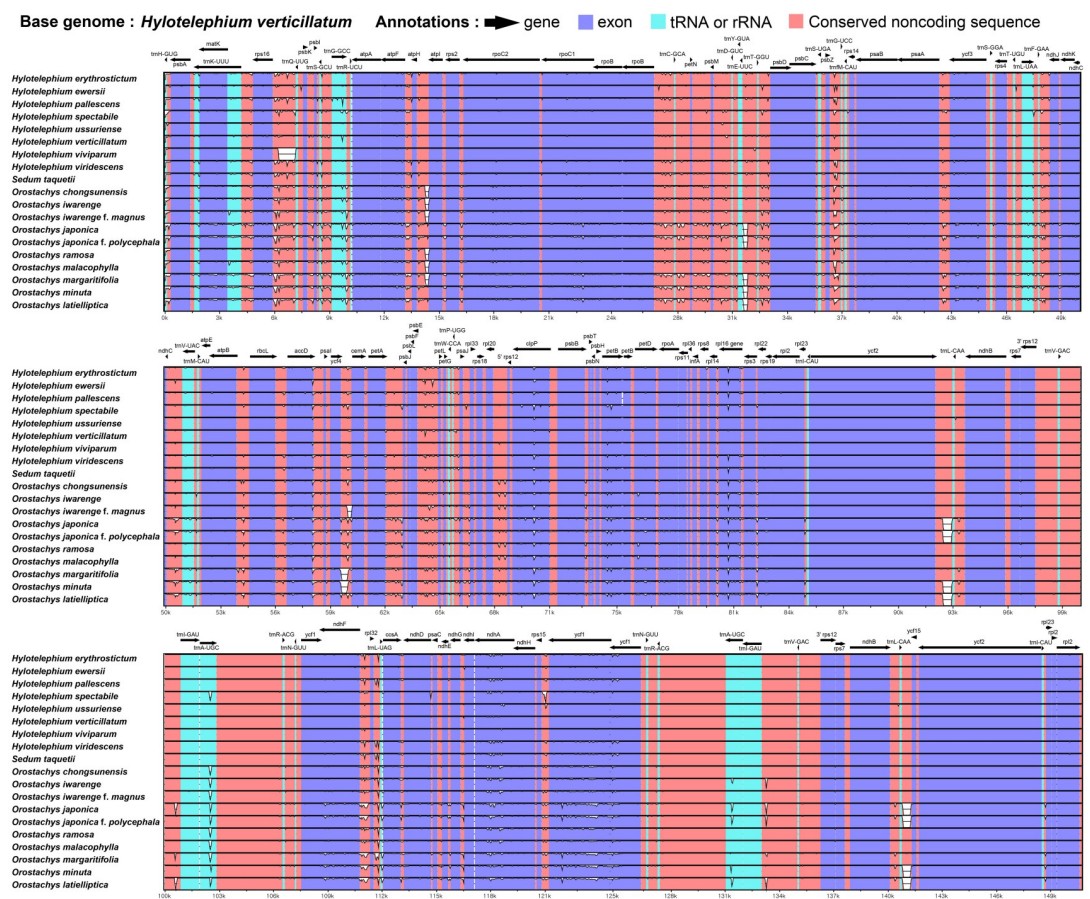

**Fig 2. Visualization of the alignment of nineteen chloroplast genomes using *H. verticilatum* as a reference.** The vertical scale indicates the percent identity, ranging from 50% to 100%. Coding genes, RNAs, and noncoding regions are marked in purple, sky blue, and red, respectively.

nineteen species spanned the LSC and IRb, IRb and SSC, and SSC and IRa regions, respectively. At the junction of IRa/LSC (JLA), *trnH-GUG* and part of the *rps19* genes were in the LCS and IRa regions, respectively. The junctions of LSC/IRb (JLB), IRb/SSC (JSB) and JLA were highly conserved with no contraction or expansion, whereas a small length change was identified in the SSC/IRa (JSA) junction. The length of *ycf1* in the IRa region was the same, but *ycf1* in the SSC region varied from 4065 to 4074 bp.

## Repeats and SSR analyses

The four types of repeat structures, including forward, reverse, complement and palindromic repeats, were identified using REPuter software [41] in the nine *Hylotelephium* chloroplast genomes. Overall, 18 (*H. ewersii*, *H. verticillatum* and *H. viviparum*) to 22 (*H. pallescens*) repeat sequences were identified in each chloroplast genome, of which 7 to 8 were forward repeats and 10 to 12 were palindromic repeats. Additionally, reverse and complement repeats were present only in *H. ewersii* (1 repeat) and *H. pallescens* (2 repeats), respectively (Fig 4A). The length of repeats ranged from 30 to 48 bp, and a repeat with a length of 41 bp was the most abundant, followed by those with lengths of 31, 37 and 30 bp (Fig 4B).

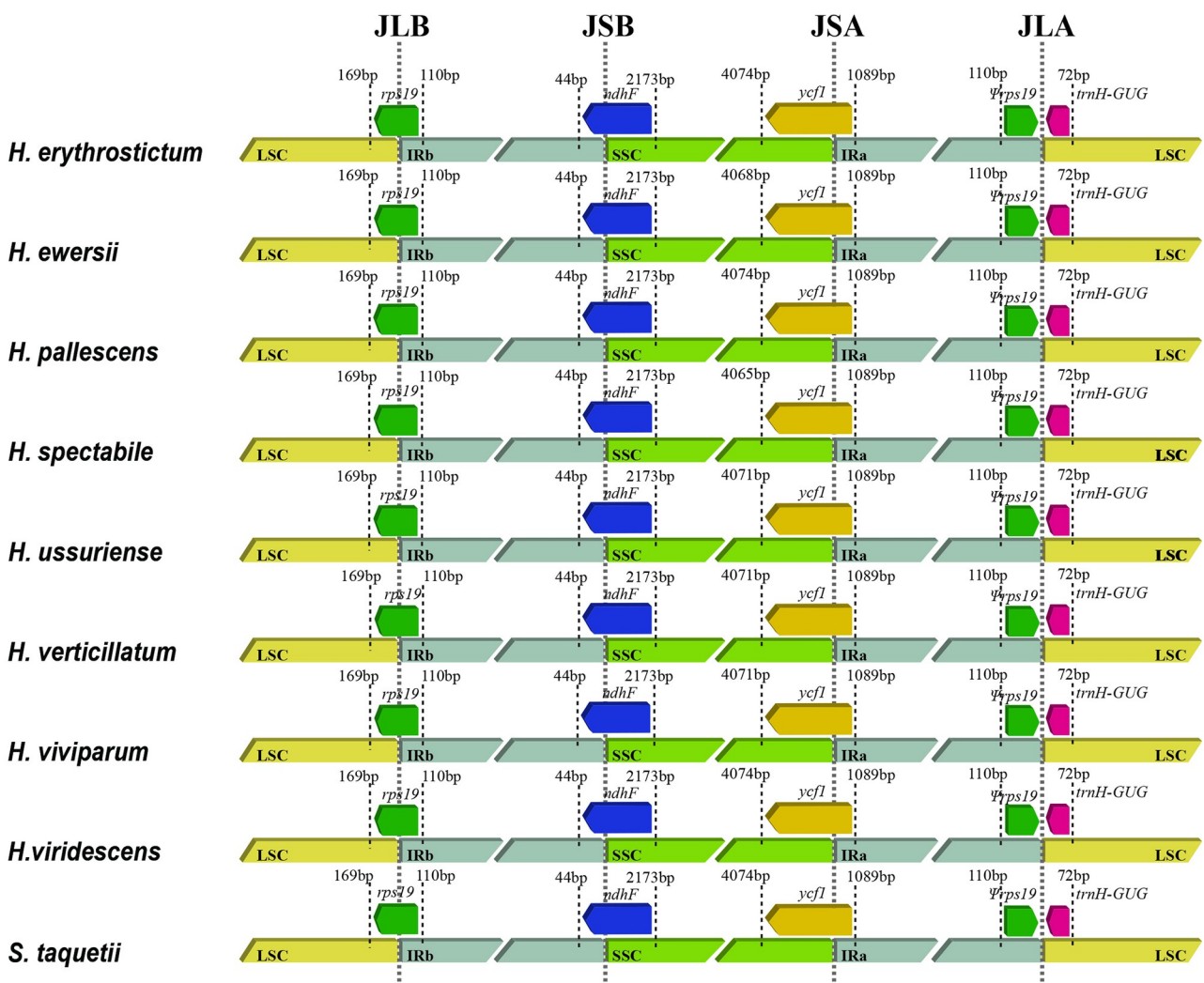

**Fig 3. Comparison of the LSC, IR, and SSC junction positions in the nineteen *Hylotelephium* and *Orostachys* chloroplast genomes.**

A total of 41 (*H. erythrostictum*, *H. pallescens*, *H. viridescens* and *S. taquetii*) to 57 (*H. specta-bile*) SSRs (also called microsatellites) were examined for the *Hylotelephium* species and ranged in size from 10 to 16 bp. All nine chloroplast genomes had no hexanucleotide repeats, and four species (*H. erythrostictum*, *H. spectabile*, *H. ussuriense*, and *H. verticillatum*) had five types of SSRs, i.e., mono-, di-, tri-, tetra- and pentanucleotides. Mononucleotide repeats ranged from 27 (*H. erythrostictum*) to 42 (*H. spectabile*) and were the most abundant in the nine *Hylotele-phium* chloroplast genomes, with A/T repeats being the only represented repeats (Fig 4C).

## Sequence divergence and mutational hotspots

The nucleotide diversity (Pi) between the nine *Hylotelephium* species and all nineteen species is shown in Fig 5. The Pi values in the IR regions were much lower than those in the LSC and SSC regions. The average Pi values were estimated to be 0.003 and 0.007. Among the *Hylotele-phium* species, the Pi values were extremely low in all regions, and only one region (*rpl32-ccsA*)

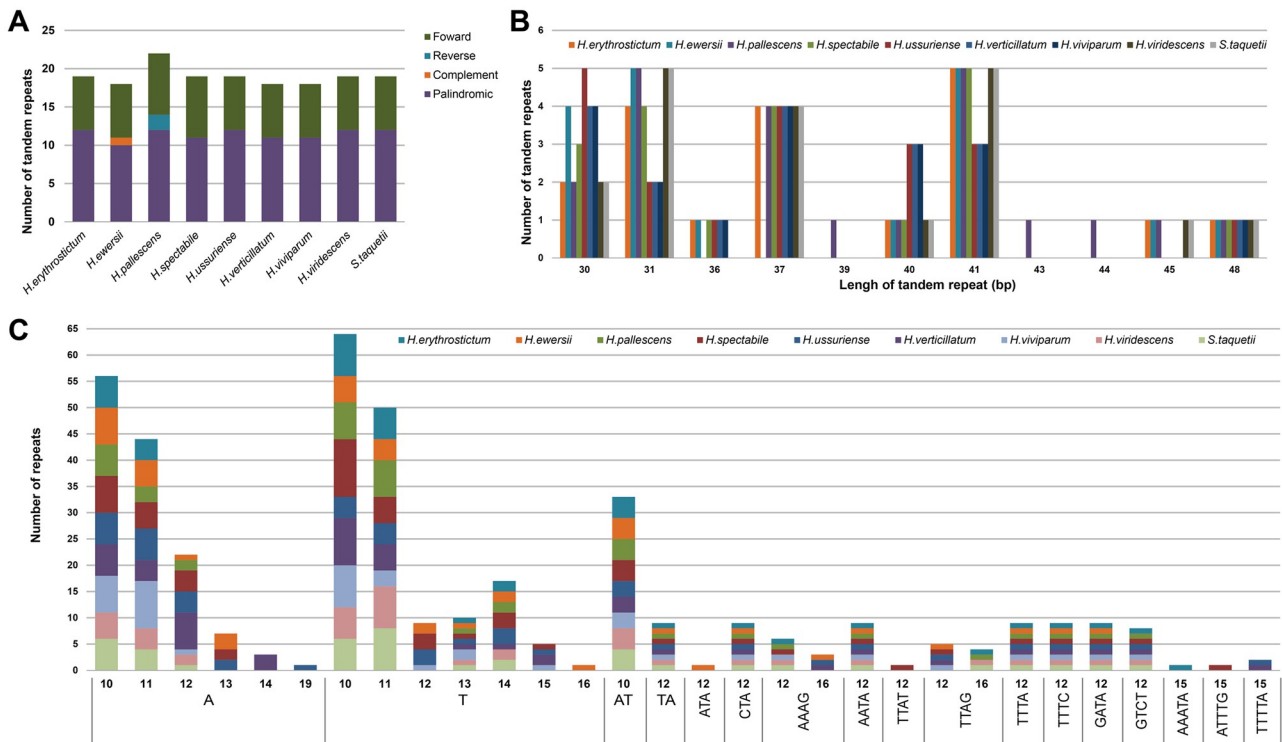

**Fig 4. Analyses of repeated sequences in the nine *Hylotelephium* chloroplast genomes.** (A) Types and number of repeats in the nine chloroplast genomes, (B) frequency by length of repeats in the nine chloroplast genomes, (C) frequency by type of SSRs in the nine chloroplast genomes.

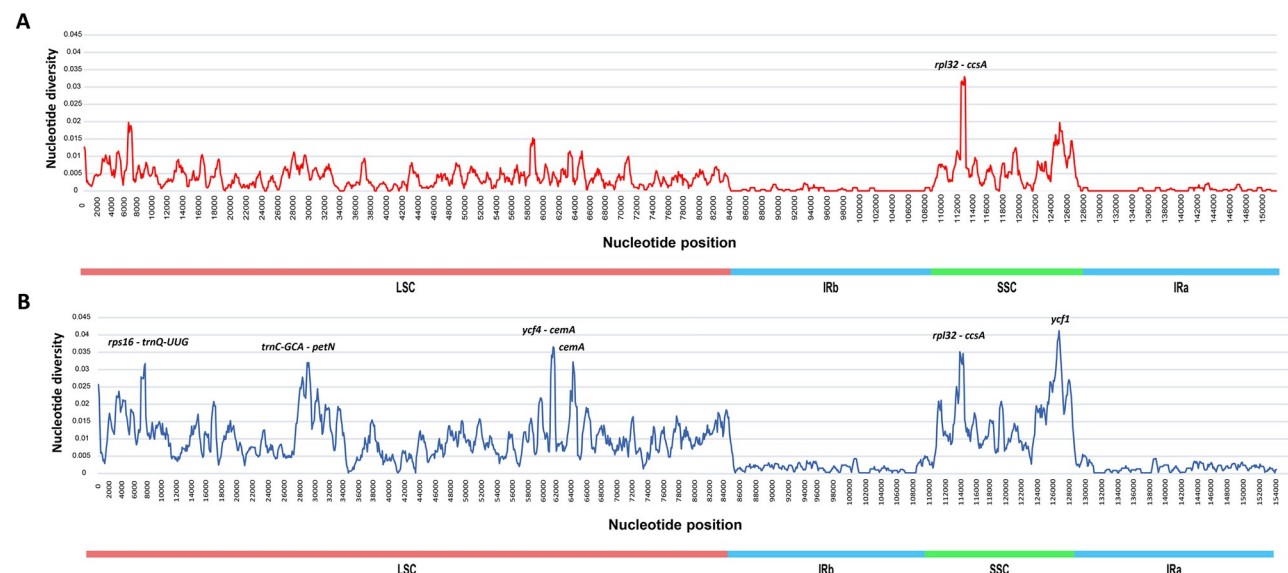

**Fig 5. Sliding window analysis of nineteen *Hylotelephium* and *Orostachys* chloroplast genomes.** (A) Pi values of nine *Hylotelephium* species and (B) Pi values of all nineteen species.

showed relatively high Pi value (>0.03). In all nineteen species, six regions had a high Pi value (>0.03). Among these, four (*rps16-trnQ*, *trnC-petN*, *ycf4-cemA*, and *cemA*) and two (*rpl32-ccsA* and *ycf1*) were in the LSC and SSC regions, respectively. Moreover, corebarcode regions, *rbcL* and *matK*, and molecular markers used in previous studies, *rps16*, and *trnL-trnF* [17–20], of nine *Hylotelephium* species and all nineteen *Hylotelephium* and *Orostachys* species pad very low Pi values of 0.0074 and 0.0176 or less, respectively (S5 Table).

## Phylogenetic analyses

The ML tree constructed with whole chloroplast genome sequences was well supported at the genus level, except for those of *Hylotelephium* and *Orostachys*. The ML tree was divided into two subclades. In the first clade, the genera *Rhodiola* L. and *Phedimus* Raf. were clustered into a well-supported, monophyletic clade with high bootstrap (BP) and Bayesian posterior probability (PP) values. The second clade consisted of *Hylotelephium*, *Orostachys*, *Meterostachys*, *Sinocrassula* A. Berger and *Umbilicus* DC. *Umbilicus* was the earliest-diverging lineage, and *Sinocrassula* was sister to *Hylotelephium*, *Orostachys* and *Meterostachys*. *Hylotelephium* was paraphyletic and formed a clade with subsect. *Orostachys* of *Orostachys* (Fig 6).

## Discussion

### Comparison of the chloroplast genomes

In this study, we collected six species of *Hylotelephium* and obtained their complete chloroplast genome sequences. The chloroplast genome of terrestrial plants is highly conserved in its nucleotide sequence and its gene content and order [47–52]. However, the gene orders of the chloroplast genome are sometimes rearranged in independent plant groups [26, 49, 50, 53, 54], and its structural rearrangements provide important systematic data. We found that the genome structure, gene content and gene order of nine *Hylotelephium* species, including the six species analyzed in this study and three published species, were identical, and the sequence identity was also very similar between species in most of the chloroplast regions (Fig 2 and S1 Fig). Therefore, we believe that the chloroplast genomes of this group are very conserved. In the *Hylotelephium* species, the genome size differed by less than 500 bp except for *H. viviparum* (Table 1). The chloroplast genome of *H. viviparum* was approximately 1000 bp shorter than that of other *Hylotelephium* species (Table 1), which was confirmed because most of the sequences of the intergenic spacer between *rps16-trnQ(UUG)* were deleted. It is considered a relatively large event in the highly conserved *Hylotelephium* chloroplast genome, and it is speculated that it can be used as a specific marker region that distinguishes *H. viviparum* from other species.

In many previous studies analyzing codon usage bias, it was confirmed that isoleucine and cysteine are the most common and the least common codons, respectively [55–58], and most codons showed higher A/T preference in the third codon [55, 59–62], probably because of the A or T abundance in the IR region [63]. As a result of this study, it was confirmed that the chloroplast genomes of *Hylotelephium* species had the same characteristics as those of general higher land plants.

The contraction and expansion of IR regions during evolution is a relatively common occurrence and has been employed as an evolutionary locus for phylogenetic studies [64–66]. In the chloroplast genome of *Hylotelephium* and *Orostachys*, however, there is little change in the IR regions (Figs 2 and 5B), so it will not be suitable for phylogenetic studies.

The number of repeats in the nine *Hylotelephium* species ranged from 18 (*H. ewersii*, *H. verticillatum* and *H. viviparum*) to 22 (*H. pallescens*), and the number of repeats according to type and length showed slight differences between species (Fig 5A and 5B). The presence and

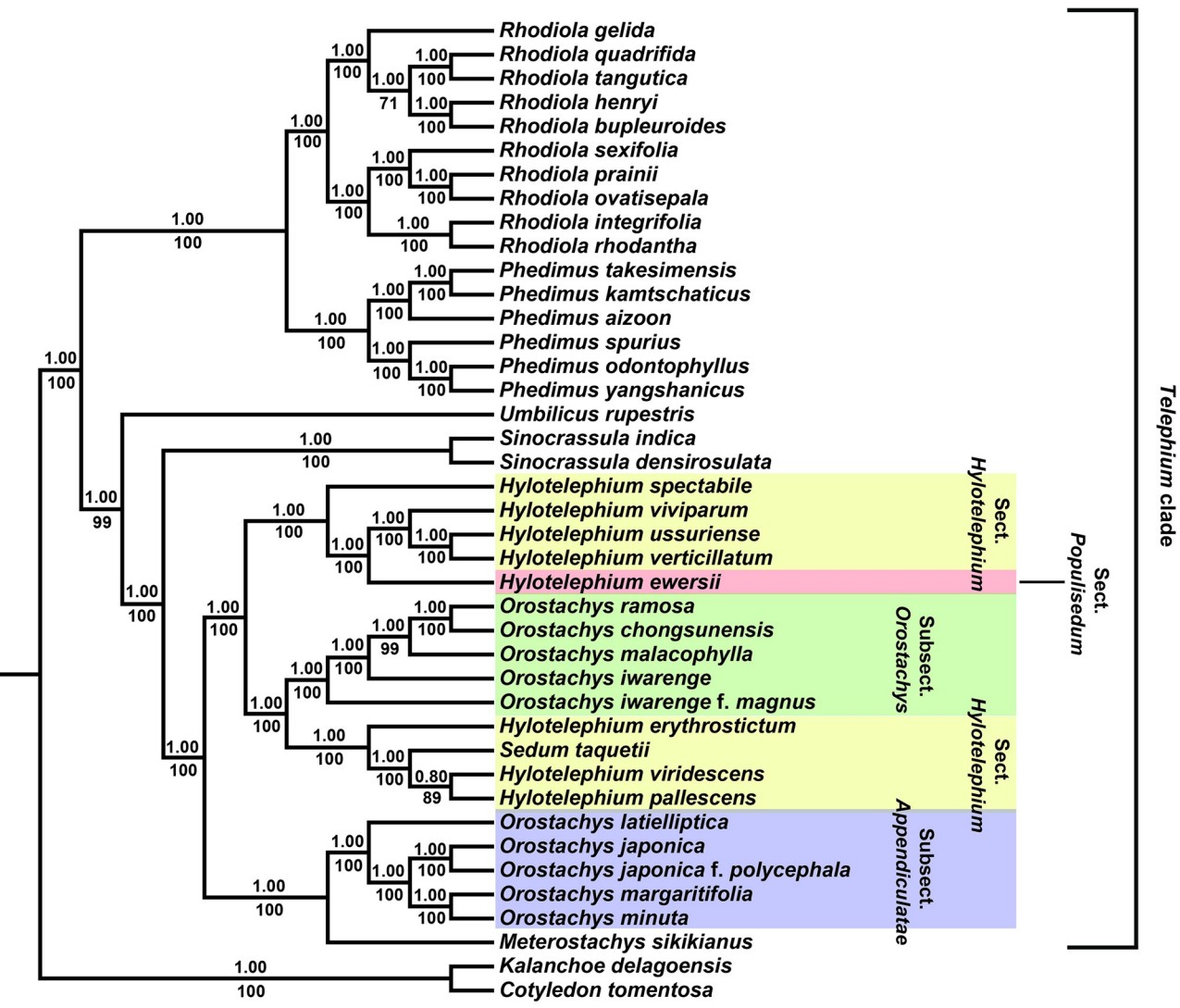

**Fig 6. The ML tree from 39 Telephium clade species and two out groups.** Bootstrap (BP) values greater than 50% are below the clades, and Bayesian posterior probabilities (PP) are shown above the clades.

abundance of repetitive sequences in the chloroplast or nuclear genome are likely to involve many phylogenetic signals [67–69], and may, therefore, provide additional evolutionary information. In addition, the SSRs identified in this study may provide various markers for population genetic studies of *Hylotelephium* species because SSRs are considered to play an important role in population genetics [58, 70].

## Selection of useful molecular marker regions for phylogeny

*Hylotelephium* is known to be a very difficult group to classify due to the high external morphological variation of each taxon and the very similar morphological characteristics between species [15, 24]. According to these findings, many morphological and molecular phylogenetic studies have been conducted, but the delimitation of this genus and phylogenetic relationships between its species remain insufficiently resolved [16–23, 71].

Previous studies have mostly used the ITS region of nuclear DNA and *matK*, *rps16*, *psbA-trnH* and *trnL-trnF* of the regions in the chloroplast genome for phylogenetic analyses [16–20]. Additionally, the CBOL Plant Working Group has recommended *matK* and *rbcL* genes as core plant barcodes [72]. The Pi values of these regions were calculated in this study, and all showed a very low Pi of 0.0074 (*matK*) or less in *Hylotelephium* chloroplast genomes and 0.0176 (*matK*) or less in *Hylotelephium* and *Orostachys* species. Therefore, the low phylogenetic resolution of the previous studies was due to the selection of molecular marker regions with very low Pi values. Additionally, it may be challenging to obtain high resolution even in the core barcode regions (*matK* and *rbcL*).

The results of this study showed that only one region (*rpl32-ccsA*) had a high Pi value (>0.03) in *Hylotelephium* species, and three regions (*rps16-trnQ*, *accD* and *ycf1*) had relatively high Pi values (>0.015) (Fig 5A). Therefore, it is considered the most suitable region to evaluate the phylogenetic relationships between *Hylotelephium* species. Furthermore, six regions (*rps16-trnQ*, *trnC-petN*, *ycf4-cemA*, *cemA*, *rpl32-ccsA* and *ycf1*) were the most suitable chloroplast regions for resolving the unclear phylogenetic relationships between *Hylotelephium* and *Orostachys* due to their high Pi values (Fig 5B).

### Assessment of phylogenetic relationships

Ohba [14] recognized two sections, *Hylotelephium* H. Ohba and *Populisedum* (A. Berger) H. Ohba, in the genus *Hylotelephium* based mainly on the insertion point of the flowering stem. He then divided the genus into three sections by treating *Sieboldia*, which was classified as a series of sect. *Hylotelephium*, as its own section [73]. *Hylotelephium* formed a polytomy or was polyphyletic with subsect. *Orostachys* of the genus *Orostachys* in many previous phylogenetic studies based on nrITS [16, 18] and some chloroplast markers [19]. Unfortunately, the genus delimitation could not be clearly observed even in this study because *Hylotelephium* formed a paraphyly with subsect. *Orostachys* of the genus *Orostachys*. We do not believe these results can be interpreted as a conclusion that the two genera should be combined into one genus because the morphological characteristics such as radical leaves and inflorescences, between the two genera greatly differ. Further studies are needed that include various species and nuclear DNA to clarify the delimitation of the genus. Additionally, the classification system below the genus level of *Hylotelephium* will also need to be reconsidered in future studies because the taxa belonging to sect. *Hylotelephium* were polyphyletic, and *H. ewersii*, which is considered in sect. *Populisedum* [18], did not form an independent clade (Fig 6).

Moreover, *Sedum taquetii*, a Korean endemic species that is distributed only on Jeju-do Island, was first described by Praeger [74] because of its larger flowers (especially carpels and petals reaching approximately 10 mm and 9 mm, respectively) compared to the taxa belonging to the same section. Since then, this species has been treated as a synonym for *H. viridescens* without taxonomic studies [13, 75], and Ohba [14] also accepted these opinions when first describing the genus *Hylotelephium* and treated it as a synonym. However, Chung and Kim [15] argued that *S. taquetii* should be recognized as an independent species because it is distinguished from *H. viridescens* in that the anther is purple. The results of this study showed that *H. viridescens* had the closest relationship with *H. pallescens*, and *S. taquetii* was the sister to the two species mentioned above. Therefore, we strongly agree with Chung and Kim [15] that *S. taquetii* should be treated as an independent taxon.

## Conclusion

In this study, we assembled the chloroplast genomes of six *Hylotelephium* species, which had a total length ranging from 150,430 to 151,717 bp. The chloroplast genomes of *Hylotelephium*

had identical structures and were highly conserved. The *Hylotelephium* species are not easy to classify because the morphological variations of each taxon are very high and there are remarkably similar morphological characteristics between species. Therefore, the four regions (*rpl32-ccsA*, *rps16-trnQ*, *accD* and *ycf1*) and six regions (*rps16-trnQ*, *trnC-petN*, *ycf4-cemA*, *cemA*, *rpl32-ccsA* and *ycf1*) presented in this study will presumably be useful for resolving the many unclear phylogenetic relationships between *Hylotelephium* species and between *Hylotelephium* and *Orostachys*, respectively. The results of the phylogenetic analysis of this study do not resolve the unclear relationships of the *Hylotelephium* species, so additional studies are needed. Furthermore, the results supported the taxonomic position of *S. taquetii*, which was treated as a synonym of *H. viridescens* in previous studies, as an independent taxon.

## Supporting information

**S1 Fig. Comparison of nine *Hylotelephium* chloroplast genome structures using the MAUVE program.**
(JPG)

**S1 Table. Information for sample collection sites, voucher specimens, sequencing results and genome assembly.**
(XLSX)

**S2 Table. The list and GenBank accession numbers used for phylogenetic analyses in this study.**
(XLSX)

**S3 Table. List of genes in the chloroplast genomes of nine *Hylotelephium* species.**
(DOCX)

**S4 Table. Codon usage bias and relative synonymous codon usage (RSCU).**
(XLSX)

**S5 Table. Pi values of four genetic regions, including corebarcode regions or molecular markers used in previous studies.**
(XLSX)

## Author Contributions

**Conceptualization:** Yoo-Jung Park, Kyeong-Sik Cheon, Kyung-Ah Kim.

**Data curation:** Sung-Mo An, Bo-Yun Kim.

**Formal analysis:** Bo-Yun Kim, Halam Kang, Ha-Rim Lee, Yoo-Bin Lee, Yoo-Jung Park, Kyung-Ah Kim.

**Funding acquisition:** Kyeong-Sik Cheon.

**Investigation:** Sung-Mo An, Halam Kang, Ha-Rim Lee, Yoo-Bin Lee, Yoo-Jung Park, Kyeong-Sik Cheon, Kyung-Ah Kim.

**Project administration:** Kyeong-Sik Cheon.

**Software:** Bo-Yun Kim, Halam Kang, Ha-Rim Lee, Kyung-Ah Kim.

**Supervision:** Kyeong-Sik Cheon.

**Validation:** Sung-Mo An.

**Visualization:** Bo-Yun Kim, Halam Kang, Ha-Rim Lee, Yoo-Bin Lee, Yoo-Jung Park, Kyung-Ah Kim.

**Writing – original draft:** Sung-Mo An, Bo-Yun Kim, Kyeong-Sik Cheon, Kyung-Ah Kim.

**Writing – review & editing:** Kyeong-Sik Cheon, Kyung-Ah Kim.

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
