## [Decision Letter · Decision Letter 0]

14 Feb 2023

PONE-D-22-30339The complete chloroplast genome sequences of six Hylotelephium species: comparative genomic analysis and phylogenetic relationshipsPLOS ONE

Dear Dr. Cheon,

Thank you for submitting your manuscript to PLOS ONE. After careful consideration, we feel that it has merit but does not fully meet PLOS ONE’s publication criteria as it currently stands. Therefore, we invite you to submit a revised version of the manuscript that addresses the points raised during the review process.

We look forward to receiving your revised manuscript.

Kind regards,

Muniyandi Nagarajan

Academic Editor

PLOS ONE

Journal Requirements:

Reviewers' comments:

Reviewer's Responses to Questions

**Comments to the Author**

1. Is the manuscript technically sound, and do the data support the conclusions?

Reviewer #1: Partly

Reviewer #2: Partly

Reviewer #3: Yes

2. Has the statistical analysis been performed appropriately and rigorously? 

Reviewer #1: Yes

Reviewer #2: N/A

Reviewer #3: Yes

3. Have the authors made all data underlying the findings in their manuscript fully available?

Reviewer #1: Yes

Reviewer #2: Yes

Reviewer #3: Yes

4. Is the manuscript presented in an intelligible fashion and written in standard English?

Reviewer #1: Yes

Reviewer #2: No

Reviewer #3: No

5. Review Comments to the Author

Reviewer #1: Chloroplast is a maternal genetic marker. Plant classification is mainly based on morphological characteristics, which is mostly consistent with the phylogenetic results of nuclear genetic markers inherited by parents. In plants, the phylogenetic results of chloroplast genetic markers and nuclear genetic markers are often different. Therefore, the introduction of the author in the introduction and discussion should clearly state which kind of marker the previous research results are based on. Moreover, based on the above reasons, the results of this study are not enough to determine the systematic location of H. viridescens, but it does provide useful information. The author may supplement the phylogenetic results of nuclear genetic markers, or please explain carefully in the results and discussion section.

Reviewer #2: Dear Authors,

Thank you for submitting your manuscript entitled “The complete chloroplast genome sequences of six Hylotelephium species: comparative genomic analysis and phylogenetic relationships”. Upon the initial review, I have several comments that will hopefully improve the paper.

1. In the data availability statement, the authors didn’t mention the accession numbers and the database wherein the genomes were deposited.

2. In the paper’s abstract, include a statement or two about the objective of the study.

3. The authors need to establish the ‘necessity’ to sequence the chloroplast genome of Hylotelephium species. The introduction part of the paper was not able to establish this clearly due to some confusing and awkward sentence construction. Kindly check the grammar and coherence of the statements. Transition sentences must be included when shifting from one point to another.

4. Please provide a detailed methodology. Include information like the library preparation method and the library insert size. The detailed process of the assembly method should be included in the paper (referenced base or de novo assembly method).

5. The chloroplast genomes’ description should include general details such as the number of duplicated genes, the number of trans-spliced genes, and the GC content of each region (LSC, SSC, IR).

6. It was stated in the paper that the cp genomes of Hylotelephium species are highly identical except for the rps16-trn-UUG. Did the authors try analyzing the insertions, deletions, and synonymous and non-synonymous substitutions? These might depict the differences among the chloroplast genome sequences.

7. Additional analyses are needed to compare the chloroplast genome sequences (e.g., SNPs analysis, relative codon usage analysis, etc.).

8. The results were simply presented but not explained in the paper, specifically the IR contraction/expansion, repeat analyses, and sequence divergence analysis. The interpretations and implications of the results were not properly discussed. All results presented must be interpreted in the discussion part and the significance of conducting these analyses must be stated in the paper.

9. Supporting details are needed to explain the results or data obtained from the study. The results and discussion should go beyond characterizing the genomes.

10. It was mentioned in the paper that only one region has a significant Pi value. Where did you get the threshold for significance? Is it the same for coding and non-coding regions? It is already established that non-coding regions are highly variable hence when comparing Pi values, it is better to compare the Pi values of the coding regions and non-coding regions separately. In this way, you can check which coding regions have higher nucleotide diversity compared to others.

11. The way the authors described the IR junctions can confuse the readers. Is there any expansion or contraction? What’s the effect of contraction or expansion in the IR junctions?

12. What’s the significance of getting the number of repeats? If one sequence has this number of repeats or repeat types, what’s the implication?

13. Are there any similarities among the Hylotelephium species and Orostachys species that grouped together in one clade? Or any possible reasons behind the clustering of some Hylotelephium species and Orostachys species?

14. The figures have low resolution, the details are not visible.

Reviewer #3: In this manuscript, authors reported six chloroplast genomes of Hylotelephium species and compared thirteen chloroplast genomes from Hylotelephium and Orostachys species published in previous studies. Authors have done detail comparison and characterization of the nineteen chloroplast genomes. The study could be interesting for publication in PLOS ONE. However, the manuscript does have several issues which I think need to be addressed.

Major：

1. The aim (2) to clarify the taxonomic identities of several taxa with ambiguous taxonomic identities. It seems there is no clear answer to this question in manuscript.

2.The aim (3) to provide important information about the most suitable chloroplast molecular markers for further studies to solve unclear phylogenetic relationships of Hylotelephium. There is no clear answer to this question in manuscript.

3.The English writing of this mauscript should be improved.

Minor：

1.Fig.1 should list the six Hylotelephium species names. The resolution of Fig. 1 was low.

2.Fig. 4 was not clear.

3.Fig. 5 was not clear.

6. PLOS authors have the option to publish the peer review history of their article (what does this mean?). If published, this will include your full peer review and any attached files.

Reviewer #1: No

Reviewer #2: No

Reviewer #3: No

---

## [Author Response · Author response to Decision Letter 0]

26 Jun 2023

We are pleased to resubmit for publication the revised version of PONE-D-22-30339 “The complete chloroplast genome sequences of six Hylotelephium species: comparative genomic analysis and phylogenetic relationships” We appreciated the constructive criticisms of the reviewers. We have addressed each of their concerns as outlined below.

→ We checked and reflected the template style of PLOS ONE at the request of the journal.

→ We have added the data availability statement in the manuscript.

Reviewer 1.

Chloroplast is a maternal genetic marker. Plant classification is mainly based on morphological characteristics, which is mostly consistent with the phylogenetic results of nuclear genetic markers inherited by parents. In plants, the phylogenetic results of chloroplast genetic markers and nuclear genetic markers are often different. Therefore, the introduction of the author in the introduction and discussion should clearly state which kind of marker the previous research results are based on. Moreover, based on the above reasons, the results of this study are not enough to determine the systematic location of H. viridescens, but it does provide useful information. The author may supplement the phylogenetic results of nuclear genetic markers, or please explain carefully in the results and discussion section.

→ Thank you very much for your constructive comments.

→ We fully agree your opinion what ‘plant classification is mainly based on morphological characteristics, which is mostly consistent with the phylogenetic results of nuclear markers inherited by parents’.

→ We have revised the Introduction and Discussion sections based on what you pointed out. Thank you again.

 

Reviewer 2.

Thank you for submitting your manuscript entitled “The complete chloroplast genome sequences of six Hylotelephium species: comparative genomic analysis and phylogenetic relationships”. Upon the initial review, I have several comments that will hopefully improve the paper.

1. In the data availability statement, the authors didn’t mention the accession numbers and the database wherein the genomes were deposited.

→ We have deposited the genome sequences of six Hylotelephium species in the GenBank of NCBI and provided the GenBank accession numbers in Table 1 and S1 Table.

2. In the paper’s abstract, include a statement or two about the objective of the study.

→ We have added it in the Abstract section.

3. The authors need to establish the ‘necessity’ to sequence the chloroplast genome of Hylotelephium species. The introduction part of the paper was not able to establish this clearly due to some confusing and awkward sentence construction. Kindly check the grammar and coherence of the statements. Transition sentences must be included when shifting from one point to another.

→ We have revised the Introduction section to emphasize the necessity of this study.

4. Please provide a detailed methodology. Include information like the library preparation method and the library insert size. The detailed process of the assembly method should be included in the paper (referenced base or de novo assembly method).

→ We have added and revised it.

5. The chloroplast genomes’ description should include general details such as the number of duplicated genes, the number of trans-spliced genes, and the GC content of each region (LSC, SSC, IR).

→ We have added it.

6. It was stated in the paper that the cp genomes of Hylotelephium species are highly identical except for the rps16-trn-UUG. Did the authors try analyzing the insertions, deletions, and synonymous and non-synonymous substitutions? These might depict the differences among the chloroplast genome sequences.

→ It is only a description of mVISTA analysis results. Also, we carried out the sequence comparison analyses in sequence divergence and mutational hotspots section of this paper.

7. Additional analyses are needed to compare the chloroplast genome sequences (e.g., SNPs analysis, relative codon usage analysis, etc.).

→ We have added the relative codon usage analysis.

8. The results were simply presented but not explained in the paper, specifically the IR contraction/expansion, repeat analyses, and sequence divergence analysis. The interpretations and implications of the results were not properly discussed. All results presented must be interpreted in the discussion part and the significance of conducting these analyses must be stated in the paper.

→ We have added some sentences.

9. Supporting details are needed to explain the results or data obtained from the study. The results and discussion should go beyond characterizing the genomes.

→ We fully agree your opinion and have done my best to do so.

10. It was mentioned in the paper that only one region has a significant Pi value. Where did you get the threshold for significance? Is it the same for coding and non-coding regions? It is already established that non-coding regions are highly variable hence when comparing Pi values, it is better to compare the Pi values of the coding regions and non-coding regions separately. In this way, you can check which coding regions have higher nucleotide diversity compared to others.

→ It means relatively high, and we have added the word ‘relatively’ to the manuscript

→ Also, what you said is a very good method, but we think the sliding window method is also a good method. In particular, we think that the biggest advantage of sliding window method is that it directly selects marker regions suitable for future phylogenetic studies by analyzing the length that can be analyzed by Sanger sequencing. because of this advantage, we analyzed using the sliding window method in this study.

→ We ask for your kind understanding.

11. The way the authors described the IR junctions can confuse the readers. Is there any expansion or contraction? What’s the effect of contraction or expansion in the IR junctions?

→ We have revised it.

12. What’s the significance of getting the number of repeats? If one sequence has this number of repeats or repeat types, what’s the implication?

→ Repeat sequences are considered important information in population genetics. So we analyzed SSR to provide information for future population genetics studies of Hylotelephium. Also, we have added it in the manuscript.

13. Are there any similarities among the Hylotelephium species and Orostachys species that grouped together in one clade? Or any possible reasons behind the clustering of some Hylotelephium species and Orostachys species?

→ We think it is because of the limitation of the resolution of the cp genome in this group. In our next paper, which is being prepared based on single copy gene sequences of nuclear genome, each section of Hylotelephium and Orostachys formed independent monophyletic group and were perfectly separated.

14. The figures have low resolution, the details are not visible.

→ We have modified the resolution of all figures to 300 dpi.

 

Reviewer 3.

In this manuscript, authors reported six chloroplast genomes of Hylotelephium species and compared thirteen chloroplast genomes from Hylotelephium and Orostachys species published in previous studies. Authors have done detail comparison and characterization of the nineteen chloroplast genomes. The study could be interesting for publication in PLOS ONE. However, the manuscript does have several issues which I think need to be addressed.

Major：

1. The aim (2) to clarify the taxonomic identities of several taxa with ambiguous taxonomic identities. It seems there is no clear answer to this question in manuscript.

→ We have revised the aim (2).

2.The aim (3) to provide important information about the most suitable chloroplast molecular markers for further studies to solve unclear phylogenetic relationships of Hylotelephium. There is no clear answer to this question in manuscript.

→ We have revised the aim (3).

3.The English writing of this mauscript should be improved.

→ Manuscript has been completely revised by a professional English translation agency.

Minor：

1.Fig.1 should list the six Hylotelephium species names. The resolution of Fig. 1 was low.

2.Fig. 4 was not clear.

3.Fig. 5 was not clear.

→ We have modified the resolution of all figures to 300 dpi and modified Figures 4 and 5.

---

## [Decision Letter · Decision Letter 1]

26 Jul 2023

PONE-D-22-30339R1The complete chloroplast genome sequences of six Hylotelephium species: comparative genomic analysis and phylogenetic relationshipsPLOS ONE

Dear Dr. Cheon,

Thank you for submitting your manuscript to PLOS ONE. After careful consideration, we feel that it has merit but does not fully meet PLOS ONE’s publication criteria as it currently stands. Therefore, we invite you to submit a revised version of the manuscript that addresses the points raised during the review process.

We look forward to receiving your revised manuscript.

Kind regards,

Muniyandi Nagarajan

Academic Editor

PLOS ONE

Reviewers' comments:

Reviewer's Responses to Questions

**Comments to the Author**

1. If the authors have adequately addressed your comments raised in a previous round of review and you feel that this manuscript is now acceptable for publication, you may indicate that here to bypass the “Comments to the Author” section, enter your conflict of interest statement in the “Confidential to Editor” section, and submit your "Accept" recommendation.

Reviewer #1: All comments have been addressed

Reviewer #2: (No Response)

Reviewer #3: All comments have been addressed

2. Is the manuscript technically sound, and do the data support the conclusions?

Reviewer #1: Yes

Reviewer #2: Yes

Reviewer #3: Yes

3. Has the statistical analysis been performed appropriately and rigorously? 

Reviewer #1: Yes

Reviewer #2: N/A

Reviewer #3: Yes

4. Have the authors made all data underlying the findings in their manuscript fully available?

Reviewer #1: Yes

Reviewer #2: Yes

Reviewer #3: Yes

5. Is the manuscript presented in an intelligible fashion and written in standard English?

Reviewer #1: Yes

Reviewer #2: No

Reviewer #3: Yes

6. Review Comments to the Author

Reviewer #1: (No Response)

Reviewer #2: Thank you for submitting the revised manuscript. I appreciate the efforts in addressing the reviewers’ comments, but upon reading the revision, I have additional comments you might consider in improving the paper.

1. The introductory parts need refinement. The description of the genus must be rephrased to fit in the paragraph, the way its phrase right now seems awkward. An awkwardly written statement can be technically correct but may sound “rough” and not proofread.

2. The third paragraph shows the disagreement on the taxonomy of Hylotelephium but the reasons behind the disagreement or differences in opinions were not properly discussed. The paragraph mentioned that “Gray [9] recognized this genus as a section of Sedum, and many taxonomists [2,10-12] agreed with Gray’s opinions, but Clausen [13] disagreed with this and classified it as a subgenus of Sedum” but there’s no follow-up statement on the reasons behind these differences in taxonomic views. What could be the morphological bases of their claims?

3. In the methodology part, kindly include the library insert size and library preparation method.

4. The result of mVista alignment needs further discussion. The statement “The LSC and SSC regions were more variable than the IR regions” does not fully interpret the result.

5. The implications of the following results are still unclear in the discussion part. How can you explain the following results?

a. High GC content in the IR regions

b. reverse and complement repeats were only present only in H. ewersii (1 repeat) and H. pallescens (2

repeats), respectively

c. Mononucleotide repeats were the most abundant, with A/T repeats being the only represented repeats

6. As indicated in the discussion part “Most of the sequences of the intergenic spacer between rps16-trnQ(UUG) were deleted. It is considered a relatively large event in the highly conserved Hylotelephium chloroplast genome”. But if deletion events are relatively important in sequence variation analysis, why not consider performing additional analysis that will highlight the mutational hotspots or nucleotide substitutions? Analysis of the SNPs and Indels or even the ratio of nonsynonymous to synonymous nucleotide substitution rates (Ka/Ks) might give more information about the sequence divergence. For instance, the Ka/Ks ratio is used to divide genes into positive selection, neutral evolution, and purification, with a limit of one. The genes with the highest Ka/Ks variability can be used as candidate barcodes to differentiate species and in the future applied to perform phylogenetic and phylogeographic analyses.

7. In the phylogeny part, it was indicated that “We do not believe these results can be interpreted as a conclusion that the two genera should be combined into one genus because the morphological characteristics between the two genera greatly differ”, can you elaborate on this?

8. In general, the paper can be enhanced by providing an impactful and effective discussion of results. The discussion must inform readers about the larger implications of your study based on the results.

Reviewer #3: In this manuscript, authors reported six chloroplast genomes of Hylotelephium species and compared thirteen chloroplast genomes from Hylotelephium and Orostachys species published in previous studies. Authors have done detail comparison and characterization of the nineteen chloroplast genomes. The manuscript has improved many areas, such as figures solutions and English writing. I agree to accept for publication in PLOS ONE.

7. PLOS authors have the option to publish the peer review history of their article (what does this mean?). If published, this will include your full peer review and any attached files.

Reviewer #1: No

Reviewer #2: No

Reviewer #3: No

---

## [Author Response · Author response to Decision Letter 1]

4 Aug 2023

Response to Reviewers

We are pleased to resubmit for publication the revised version of PONE-D-22-30339 “The complete chloroplast genome sequences of six Hylotelephium species: comparative genomic analysis and phylogenetic relationships” We appreciated the constructive criticisms of the reviewers. We have addressed each of their concerns as outlined below.

Reviewer 1.

No Response

→ Thank you very much for agreeing to publish our paper in PLOS ONE.

Reviewer 2.

Thank you for submitting the revised manuscript. I appreciate the efforts in addressing the reviewers’ comments, but upon reading the revision, I have additional comments you might consider in improving the paper.

1. The introductory parts need refinement. The description of the genus must be rephrased to fit in the paragraph, the way its phrase right now seems awkward. An awkwardly written statement can be technically correct but may sound “rough” and not proofread.

→ We have revised it.

2. The third paragraph shows the disagreement on the taxonomy of Hylotelephium but the reasons behind the disagreement or differences in opinions were not properly discussed. The paragraph mentioned that “Gray [9] recognized this genus as a section of Sedum, and many taxonomists [2,10-12] agreed with Gray’s opinions, but Clausen [13] disagreed with this and classified it as a subgenus of Sedum” but there’s no follow-up statement on the reasons behind these differences in taxonomic views. What could be the morphological bases of their claims?

→ Unfortunately, many older taxonomic literatures, including those mentioned above, did not explained why. So, we couldn’t write about the reason either.

3. In the methodology part, kindly include the library insert size and library preparation method.

→ We have added it.

4. The result of mVista alignment needs further discussion. The statement “The LSC and SSC regions were more variable than the IR regions” does not fully interpret the result.

→ We have revised and added it.

5. The implications of the following results are still unclear in the discussion part. How can you explain the following results?

a. High GC content in the IR regions

b. reverse and complement repeats were only present only in H. ewersii (1 repeat) and H. pallescens (2 repeats), respectively

c. Mononucleotide repeats were the most abundant, with A/T repeats being the only represented repeats

→ They are general descriptions of the results of the analysis of cp genome of this group.

→ It would be nice if all the results were connected to the discussion, but we think there are times when it is not necessary or not possible to do so.

6. As indicated in the discussion part “Most of the sequences of the intergenic spacer between rps16-trnQ(UUG) were deleted. It is considered a relatively large event in the highly conserved Hylotelephium chloroplast genome”. But if deletion events are relatively important in sequence variation analysis, why not consider performing additional analysis that will highlight the mutational hotspots or nucleotide substitutions? Analysis of the SNPs and Indels or even the ratio of nonsynonymous to synonymous nucleotide substitution rates (Ka/Ks) might give more information about the sequence divergence. For instance, the Ka/Ks ratio is used to divide genes into positive selection, neutral evolution, and purification, with a limit of one. The genes with the highest Ka/Ks variability can be used as candidate barcodes to differentiate species and in the future applied to perform phylogenetic and phylogeographic analyses.

→ We fully agree your opinion.

→ However, we believe that we have performed analyzes and discussions that sufficiently explain of our study aim.

7. In the phylogeny part, it was indicated that “We do not believe these results can be interpreted as a conclusion that the two genera should be combined into one genus because the morphological characteristics between the two genera greatly differ”, can you elaborate on this?

→ We have added it.

8. In general, the paper can be enhanced by providing an impactful and effective discussion of results. The discussion must inform readers about the larger implications of your study based on the results.

→ Thank you very much for your sincere advice.

Reviewer 3.

In this manuscript, authors reported six chloroplast genomes of Hylotelephium species and compared thirteen chloroplast genomes from Hylotelephium and Orostachys species published in previous studies. Authors have done detail comparison and characterization of the nineteen chloroplast genomes. The manuscript has improved many areas, such as figures solutions and English writing. I agree to accept for publication in PLOS ONE.

→ Thank you very much for your kind comments and for agreeing to publish our paper in PLOS ONE.

---

## [Editor Report · Decision Letter 2]

13 Sep 2023

The complete chloroplast genome sequences of six Hylotelephium species: comparative genomic analysis and phylogenetic relationships

PONE-D-22-30339R2

Dear Dr. Kyung-Ah Kim,

We’re pleased to inform you that your manuscript has been judged scientifically suitable for publication and will be formally accepted for publication once it meets all outstanding technical requirements.

Kind regards,

Muniyandi Nagarajan

Academic Editor

PLOS ONE
---

## [Editor Report · Acceptance letter]

2 Oct 2023

PONE-D-22-30339R2 

The complete chloroplast genome sequences of six *Hylotelephium* species: comparative genomic analysis and phylogenetic relationships 

Dear Dr. Cheon:

I'm pleased to inform you that your manuscript has been deemed suitable for publication in PLOS ONE. Congratulations! Your manuscript is now with our production department. 

Kind regards, 

on behalf of

Dr. Muniyandi Nagarajan 

Academic Editor

PLOS ONE